# Doppler imaging detects bacterial infection of living tissue

Honggu Choi [1], Zhe Li[1], Zhen Hua[1], Jessica Zuponcic[2], Eduardo Ximenes[2], John J. Turek[3], Michael R. Ladisch [2,4] & David D. Nolte[1✉]

Living 3D in vitro tissue cultures, grown from immortalized cell lines, act as living sentinels as pathogenic bacteria invade the tissue. The infection is reported through changes in the intracellular dynamics of the sentinel cells caused by the disruption of normal cellular function by the infecting bacteria. Here, the Doppler imaging of infected sentinels shows the dynamic characteristics of infections. Invasive *Salmonella enterica* serovar Enteritidis and *Listeria monocytogenes* penetrate through multicellular tumor spheroids, while non-invasive strains of *Escherichia coli* and *Listeria innocua* remain isolated outside the cells, generating different Doppler signatures. Phase distributions caused by intracellular transport display Lévy statistics, introducing a Lévy-alpha spectroscopy of bacterial invasion. Antibiotic treatment of infected spheroids, monitored through time-dependent Doppler shifts, can distinguish drug-resistant relative to non-resistant strains. This use of intracellular Doppler spectroscopy of living tissue sentinels opens a new class of microbial assay with potential importance for studying the emergence of antibiotic resistance.

[1] Department of Physics and Astronomy, Purdue University, 525 Northwestern Ave, West Lafayette, IN 47907, USA. [2] Department of Agricultural and Biological Engineering and the Laboratory of Renewable Resources Engineering, Purdue University, West Lafayette, IN 47907, USA. [3] Department of Basic Medical Science, Purdue University, 625 Harrison St, West Lafayette, IN 47907, USA. [4] Weldon School of Biomedical Engineering, Purdue University, West Lafayette, IN 47907, USA. ✉email: nolte@purdue.edu

Pathogenicity studied at a molecular level includes host-cell identification, cellular signaling, infection, and spreading strategies of various pathogens within cell culture[1–10]. One of the principal methods to study the behavior of pathogens is fluorescence dynamic microscopy[6,7,11,12] that monitors the two-dimensional behavior of pathogens in real-time[13–15]. However, the fluorescence technique requires genetic engineering or staining to insert fluorescing agents and may not represent natural phenotypes. In addition, the number of specimens monitored by microscopy is limited by the field of view of the optics, and pathogen detection by fluorescence microscopy has limitations when pathogens are unknown. To develop a protocol that can monitor pathogens with natural 3D phenotypes, and without further processing, we introduce biodynamic imaging (BDI) of bacterial invasion of living tissue sentinels.

Biodynamic imaging[16] is based on low-coherence interferometry with coherence-gated ranging based on digital holography to record scattered light from a selected depth inside living tissue while rejecting incoherent background[10,17,18]. Dynamic speckle detected using BDI is caused by light scattering from moving intracellular components that have persistent transport that places them in the Doppler light scattering regime, based on Doppler shifts caused by dynamic light scattering[19], for which the persistence length of the transport is greater than the reduced wavelength of the probe light, expressed as $\ell_p > \lambda_0/4\pi n$, where $\ell_p$ is the mean transport persistence length, $\lambda_0 = 840$ nm is the free-space wavelength and $n \approx 1.4$ is the refractive index of the tissue. (For a detailed analysis of intracellular Doppler light scattering see ref. [19]) BDI of living tissues has been used for drug screening[20], for the assessment of chemoresistance for canine and human patients[21,22], and for oocyte and embryo viability in artificial reproductive technology[23]. In this article, we describe the first application of BDI to probe subtle changes in the intracellular motions of in vitro living host tissue induced by an early-stage bacterial infection.

Bacterial infection of living tissue and the rise of antibiotic resistance pose a serious threat to the future of human health caused by bacteria including *Streptococcus pneumoniae*, *Neisseria meningitidis*, *Staphylococcus aureus*, and *Pseudomonas aeruginosa*, among others. By the year 2050, it is projected that more people will die from septic shock, caused by antibiotic resistance, than from cancer[24]. Mismanagement of treatments and the indiscriminate use of antibiotics in livestock feed has accelerated antibiotic resistance that may soon render common antibiotics inert[25,26] Therefore, rapid assessment of infection at the early stages of septic shock, and the selection of the most effective treatment will be a pressing need[27,28]. To demonstrate the viability of BDI to characterize bacterial infection of living tissue and resistance to treatment, this study uses food pathogens, including pathogenic and non-pathogenic serovars that can be handled safely, for which the emergence of antibiotic resistance also has widespread implications.

## Results

**Experimental principles and functional imaging.** The speeds of intracellular dynamics range across three orders of magnitude from tens of nanometers per second (cell crawling, metastasis, blebbing, apoptotic bodies)[29–32] to tens of microns per second (organelles, vesicles, mitochondria)[33–36] with associated Doppler frequencies spanning from 10 mHz to 10 Hz[19]. The conversion from intracellular speed to Doppler frequency for the infrared backscatter geometry is $\Delta\omega_D = \vec{q} \cdot \vec{v}$, where $q = 4\pi n/\lambda_0$ is the momentum transfer and $v$ is the intracellular speed. The deep sub-Hertz of the slowest motions places the detection of relative Doppler frequency shifts at less than one part in $10^{16}$ Hz. Such

ultra-low-frequency Doppler shifts are detected through phase-sensitive ultra-stable digital holography combined with low-coherence infrared light scattering to capture the complex spectra of beat frequencies from dynamic speckle[37,38]. The Doppler signals are selected by coherence-gating and isolated to optical sections up to a millimeter deep inside living tissue. Multiple light scattering from moving intracellular constituents compound Doppler frequency shifts (summed in the exponential phases) to produce high-dynamic-range fluctuations corresponding to a continuum of Doppler beat frequencies centered on zero frequency (isotropic intracellular transport).

The coherence-gated optical system configuration is shown in Fig. 1a. The Mach–Zehnder interferometer matches the optical path length of the adjustable reference arm to achieve a zero-path difference relative to light scattered from a selected depth inside the tissue sample. The reference and signal waves are incident off-axis on a CMOS pixel array at a small angle to produce spatial carrier fringes on the array. The camera is on a Fourier plane of the optical system and records a Fourier-domain hologram at a frame rate of 25 fps. At the end of a 14-h experiment, the holograms are reconstructed using two-dimensional spatial Fourier transforms. Time-lapse frames of the tissue sample are converted to time series that are analyzed in the frequency domain and summed to produce fluctuation power spectra. There are several modes of BDI, including optical coherence imaging (OCI) that captures structural properties of the sample, motility contrast imaging (MCI) that maps the Doppler activity volumetrically in 3D or 4D (time-lapse), tissue-dynamics spectroscopy (TDS) that forms time-frequency spectrograms that track the spectral changes in time, and tissue-dynamics spectroscopic imaging (TDSI), which is spatially resolved TDS (see "Methods" and ref. [39]).

A key element of this technique applied to bacterial infection is the use of sub-millimeter-scale 3D living tissue sentinels that are the living matrix to support and dynamically respond to the bacterial invasion of as little as $10^3$ viable bacterial pathogens per 500 micron-diameter sentinel, or about one bacterium per 100 epithelial cells. The light scattered from infected living tissue is dominated by Doppler scattering from the cellular constituents of the tissue rather than from the bacteria itself. In this way, the sentinels not only make the action of the infection visible but also transmit the changes in intracellular dynamics caused by the infection. The reversible nature of the response of this system, for instance, when antibiotics are applied to suppress the infection and the intracellular dynamics revert to normal behavior, is a unique feature of this sentinel system.

The sentinels are three-dimensional living tissue culture grown from biologically relevant cell lines (ATCC) into multicellular spheroids. The spheroid growth is highly reproducible and can generate large replicate numbers for multi-well plate assays. The three-dimensional nature of the spheroids is a crucial characteristic because conventional two-dimensional tissue culture fails to reproduce important phenotypic properties for biologically relevant assays[40–42]. The study of infection by food-borne pathogens (*Salmonella enterica* serovar Enteritidis phage type (PT) 21, *Listeria monocytogenes*, *Listeria innocua*, and *Escherichia coli*) reported here used the DLD-1 colon adenocarcinoma cell line to grow multicellular spheroids with characteristics of epithelial colon tissue, which is a relevant target for these pathogens. The growth of the DLD-1 spheroids is described in the "Methods" section and Supplemental Information.

The *E. coli* strain used in this study is the genetically engineered O157:H7 strain for which the Shiga toxin gene has been removed, and green fluorescent protein (GFP) and ampicillin resistance genes have been added[43]. The *S. enterica* and *L. monocytogenes* strains are natural phenotypes, which can infect mammalian cells

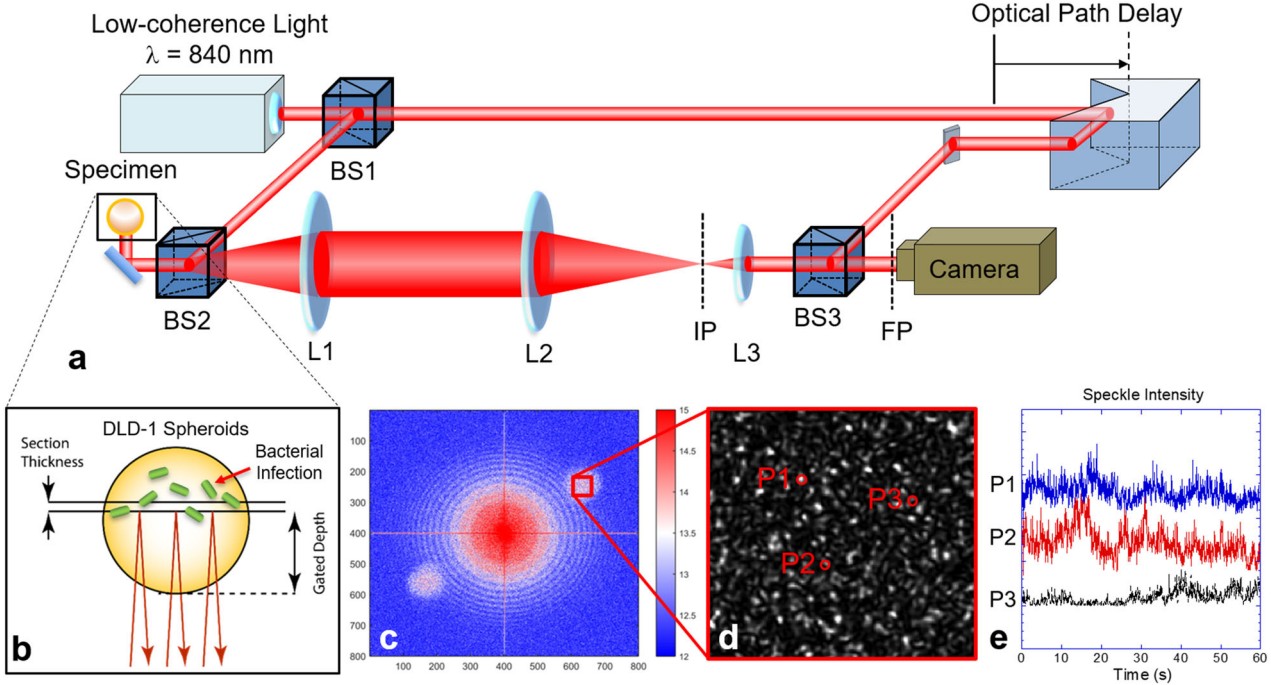

**Fig. 1 Biodynamic imaging system configuration. a** Experimental principles and setup of intracellular Doppler spectroscopic imaging. A living sample (tumor spheroid or biopsy) is optically sectioned with low-coherence backscattered infrared light on a digital camera on the Fourier plane using a Mach–Zehnder off-axis digital holographic system. The low-coherence light is split by a beam splitter (BS1) into the object and reference arms. Backscattered light is collected by the 4-*f* system (L1 and L2) and forms an image on the image plane (IP). The Fourier lens (L3) transfers the image onto the Fourier plane (FP). **b** Scattering of light from dynamic intracellular processes generates dynamic speckle that is altered when tumor spheroids are inoculated with various bacterial strains. **c** The Fourier-domain digital hologram is reconstructed to the image plane using a fast Fourier transform. The zero-order is in the center, and the two first-order side-bands are on the diagonal. The red box shows a magnified part of the image plane speckle. **d** A magnified reconstructed speckle image. **e** Dynamic speckle intensity fluctuates temporally at different locations P1, P2, and P3.

and proliferate. In the case of *S. enterica*, the bacteria infiltrate host cells by modulating the cellular membrane to induce engulfing motion causing the bacteria to be internalized. *L. monocytogenes* physically penetrate mammalian cell walls and synthesize actin filaments to gain propulsion force for cell-to-cell spreading. On the other hand, *L. innocua* is a strain of *Listeria* genus that is non-invasive and is used as a negative control in the experimental comparisons of pathogenicity.

**Biodynamic imaging of bacterial invasion into tissue.** Optical coherence images (OCI) are averaged over 2048 frames to generate the OCI maps. OCI maps of tumor spheroids are shown in Fig. 2a before and 6 h after infection of DLD-1 for the four bacterial strains *E. coli*, *S. enterica*, *L. monocytogenes*, and *L. innocua* at an exposure of 10⁷ CFU per well. The associated time courses of the average backscatter brightness (BB) are shown in Fig. 2c. Proliferating bacteria produce increasing brightness with infection. The infection by *E. coli* produces the strongest enhancement of brightness as the proliferating mass of bacteria increases the refractive index heterogeneity on the exterior to the cell bodies[44]. The weakest BB effect is from *L. innocua* that is slightly above the control level due to the inefficient proliferation rate[45]. The two pathogenic strains *L. monocytogenes* and *S. enterica* show intermediate BB increases with time.

A key biodynamic metric is the temporal speckle contrast quantified as the normalized standard deviation of the backscatter brightness that expresses the overall Doppler activity of the tissue and is displayed as motility contrast images (MCI). MCI before and 6 h after infection by the four bacteria strains are shown in Fig. 2b. The decrease in MCI under bacterial infection is a combination of increased BB and a change in Doppler activity.

The overall effect of the infection is inhibition of intracellular dynamics that may be in part from the bacteria competing for the same nutrients or related to the effects of metabolic by-products from the bacteria (Supplementary Note 2.2). The dynamic range (DR) of the spectral density is another marker that estimates the change in dynamics. The dynamic range is defined as the log ratio of the spectral amplitude at the lowest frequency (10 mHz) to the spectral amplitude at the Nyquist frequency (12.5 Hz). The amplitude at the Nyquist frequency is also called the "Nyquist floor", which is another useful dynamic marker. The Nyquist floor increases when there is an increase in high-speed transport, for instance, caused by organelle or vesicle transport. The rise of the Nyquist floor decreases the dynamic range if there is no corresponding increase of amplitude at the lowest frequency. On the other hand, the dynamic range increases if the dynamics around 0.01 Hz in the low-frequency band contains an increased fraction of intracellular components that move slowly. The DR is calculated by averaging the first 5 spectral components compared against the average of the last 5 spectral components. The changes in dynamic range relative to the pre-inoculation averages are shown in Fig. 2d. Increased dynamic range after inoculation was observed for *E. coli*, *S. enterica*, and *L. innocua*, but not for *L. monocytogenes*. Non-monotonic behavior may represent successive "waves" of infection or competition for nutrients.

The fluctuation Doppler spectra of the DLD-1 tissue prior to infection and 6 h after infection are shown in Fig. 3a for the four bacterial strains. The spectral power in the mid frequencies between 0.1 and 1 Hz is suppressed in all cases by the infection. The associated time-frequency spectrograms are shown in Fig. 3b. The spectrograms are calculated using[16]

$$D(\nu, t) = \log S(\nu, t) - \log S(\nu, t_0) \tag{1}$$

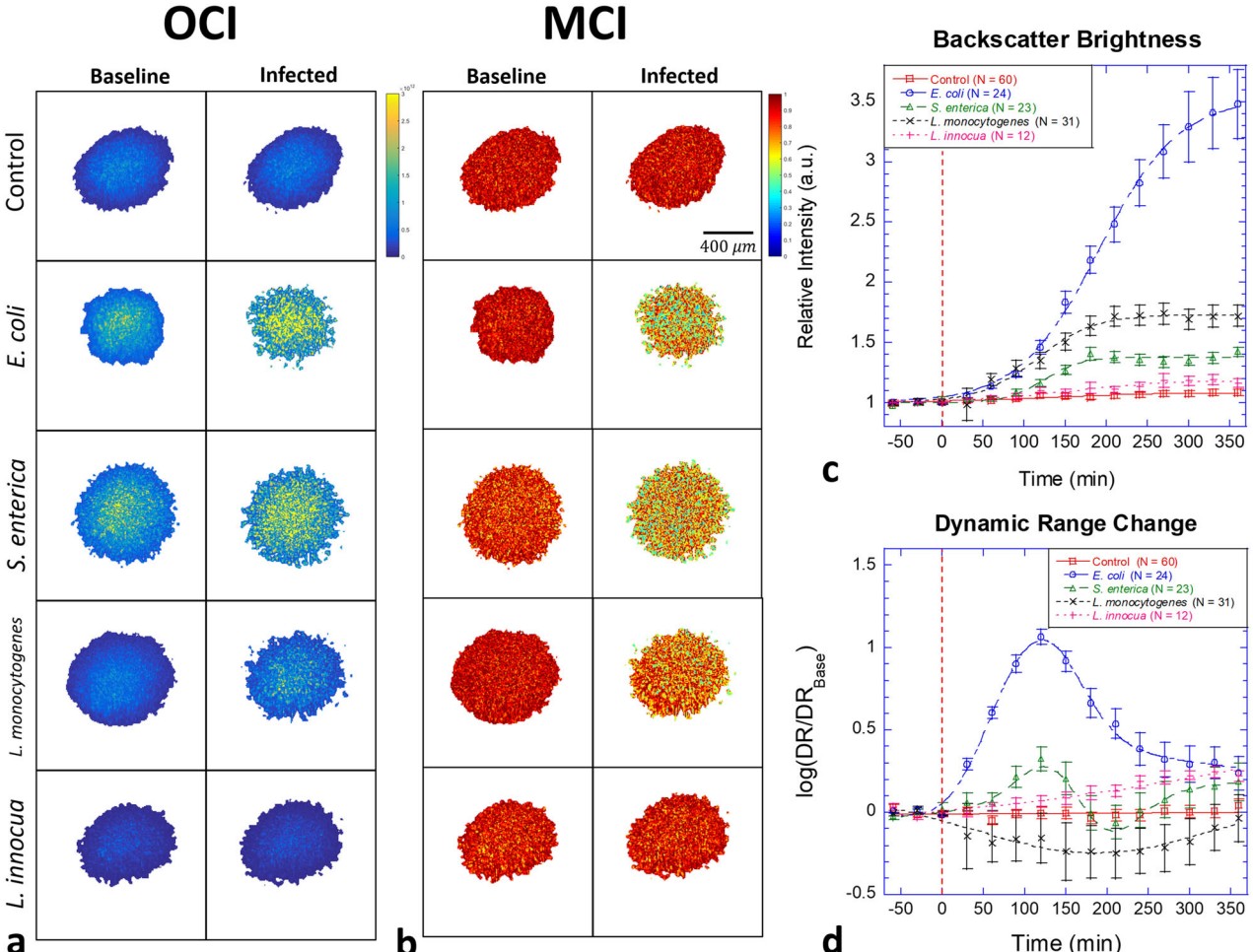

**Fig. 2 Examples of DLD-1 multicellular tumor spheroids before and after infection. a** Optical coherence images (OCI) of the optical mid-sections from DLD-1 spheroids showing backscatter brightness before (left) and 6 h after (right) infection by $10^7$ bacterial CFU per well. The strongest increase in backscatter brightness (BB) occurs for *E. coli* (note second OCI column). **b** Motility contrast images (MCI) for the same samples. The growth of the bacteria causes a strong inhibition of Doppler activity for *E. coli* (note second MCI column) and moderate inhibition of Doppler activity for *Listeria monocytogenes* and *Salmonella enterica* serovar Enteritidis. Slowly proliferating *Listeria innocua* shows the smallest effect. The scale bar applies to all speckle images. **c** Temporal evolution in BB as a function of time after inoculation (red dashed line). Infection by *E. coli* produces a large increase in BB. **d** Dynamic range (DR) of the spectral density changes non-monotonically as a function of time. The error bars represent standard error.

where $S$ is the spectral power density, $\nu$ is the Doppler frequency shift, and $t_0$ is the baseline time when the bacteria are pipetted into the well. Spectral power densities were normalized by the intensity or by zero-sum to obtain relative dynamic density and spectral shape change, respectively (Supplementary Figs. 1 and 2). The baseline spectrum of the DLD-1 tissue is established for 1.5 h prior to inoculation (infection) by the bacteria, and the relative change in the spectral power is tracked for 6 h after infection. The spectrograms for all bacteria show suppressed spectral densities (decreased intracellular dynamics), although *S. enterica* shows a slight high-frequency enhancement with weak suppression at low-frequency. The effects of bacterial infection on the tissue dynamics are extremely marked. The spectral changes induced by the bacterial infection with an initial dose of $10^6$ CFU/well are two to five times larger than is typically observed for cytostatic or cytotoxic drug effects (usually <30%) in the same cell line[46].

As shown in Fig. 3, the time evolution of the spectrograms shows different responses of DLD-1 to different bacterial infections. *E. coli* and *L. monocytogenes* cause the strongest spectral enhancement at low frequencies (the rheology band associated with cell shape changes and cell death)[35]. *S. enterica* infection shows a weaker spectral enhancement at low frequency

but displays a moderate enhancement at higher frequencies (associated with organelle transport). In addition to these mechanistic effects, growing bacteria deplete limited resources in a growth medium and increase $CO_2$ concentration that alters the pH level in the growth medium, which can perturb cellular dynamics[47] (Supplementary Table 1). Pathogenic *S. enterica* is internalized and replicates inside host cells inducing active organelle transfer for replication, which is consistent with high-frequency enhancement. The temporal responses occur slowly, and the contour lines show the trends of the spectral shifts. The standard deviations of the spectral responses over the sample replicates ($N = 68, 36, 12,$ and $13$, respectively) for all frequency bands are $\sigma_{\text{control}} = 0.01$, $\sigma_{E.\ coli} = 0.30$, $\sigma_{S.\ enterica} = 0.03$, $\sigma_{L.\ monocytogenes} = 0.08$, and $\sigma_{L.\ innocua} = 0.01$.

**Time-lapse tissue-dynamics spectroscopic imaging**. Tissue-dynamics spectroscopic imaging (TDSI) performs TDS on small groups of pixels and applies linear filters to extract dynamic spectral information such as time-dependent changes in the organelle or rheology bands (see "Methods" section)[39]. The output of the linear filters creates a map of dynamic information

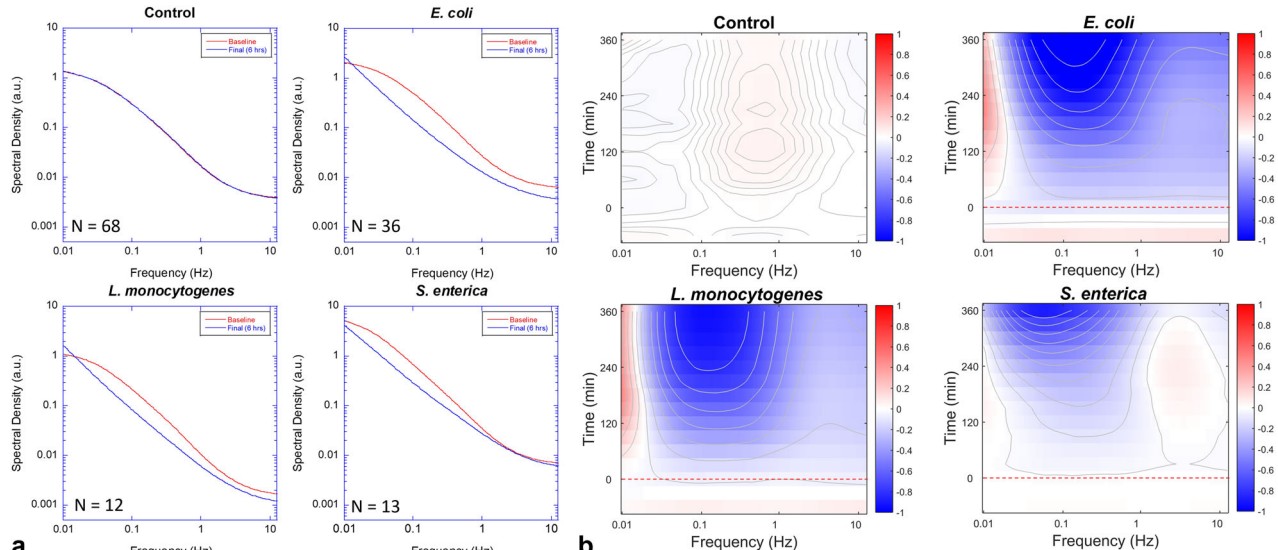

**Fig. 3 Spectral responses by various bacteria. a** Changes in the Doppler spectra (intensity normalized) for the four bacterial strains with initial loads of $10^6$ CFU per well. Mid frequencies are inhibited in all cases with the strongest effect from *E. coli*. The high-frequency Nyquist floor (related to organelle transport) increases noticeably for *S. enterica*. The values for *N* are the number of replicate samples used in the average. The uncertainty in spectral amplitudes is ±4%. **b** The associated spectrograms are generated using Eq. (2). The samples are inoculated (red dashed line) after a baseline is established. The contours (10 in each graph) help show the general trends.

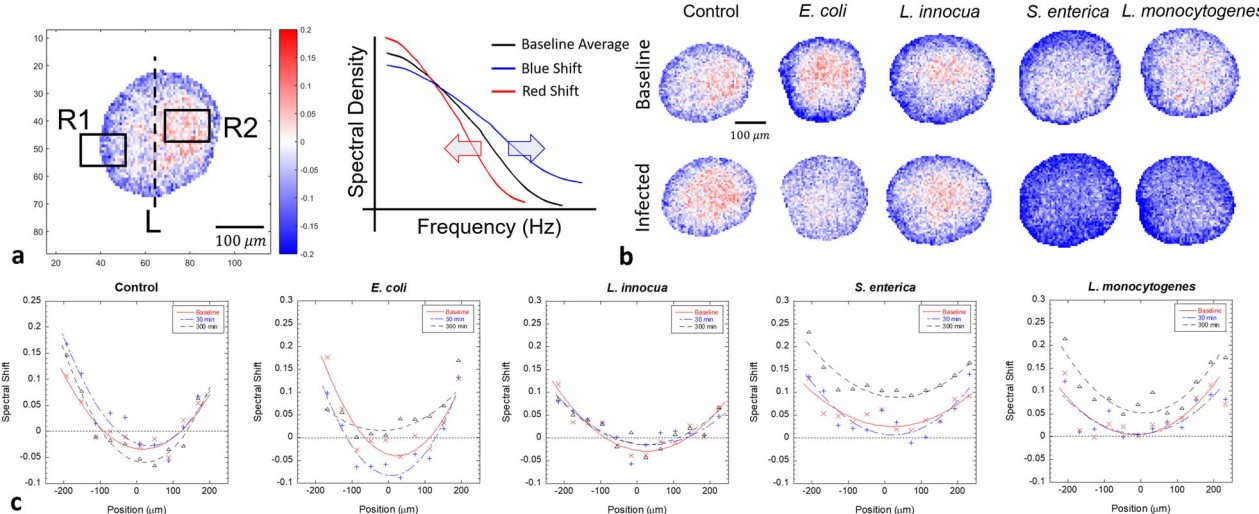

**Fig. 4 Representative cases of tissue-dynamics spectroscopic images (TDSI) of bacterial invasion of $10^7$ CFU into a DLD colon adenocarcinoma tumor spheroid. a** The linear filter used here measures red or blue shifts of spectral shape (zero-sum normalization). The untreated control displays a small blue shift (blue colored) in the outer layers (R1) and a small redshift (red-colored) in the core (R2) over the duration of the experiment (5 h). **b** *L. innocua* is indistinguishable from the control. *S. enterica* and *L. monocytogenes* display strong blue shifts throughout the volume within 300 m after infection. The blue shift for *E. coli* is intermediate between the pathogenic strains and the control. The scale bar applies to all other speckle images in **b**. **c** One-dimensional plots of TDSI along the axis L in **a**. The pathogen group (*L. monocytogenes* and *S. enterica*) display blue shifts throughout the volume. The *y* axis represents the relative change in average Doppler frequency (knee frequency of the spectrum) The two pathogenic strains induce approximately a 15% increase in intracellular speeds.

across the optical section of the sample. Spectral pattern filters track spectral shape changes, which are related to mechanistic changes in the tissue (such as a change in the average intracellular speed), and tissue-dynamics spectroscopic images (TDSI) are generated that map out these mechanistic effects across the optical section of the sample. Examples of TDSI from the infection by the bacterial strains are shown in Fig. 4. The linear filter in this example is a first-order Legendre polynomial that captures the red or blue shift of frequencies within a group of pixels. The shifts are associated with decreased or increased average

intracellular speeds, respectively. The control (Fig. 4b, top row) shows a slight blue shift in the outer regions of the tumor spheroids accompanied by a slight redshift in the core region. The core region of larger spheroids (approx. half a millimeter) is relatively hypoxic, and transport limitations can decrease nutrient concentration. Infection by (Fig. 4b, bottom row) *E. coli*, *S. enterica*, and *L. monocytogenes* produces blue shifts associated with high-speed dynamics. The two invasive strains, *S. enterica*, and *L. monocytogenes* show the strongest blue shifts that uniformly infect the entire spheroid after 5 h, consistent with the

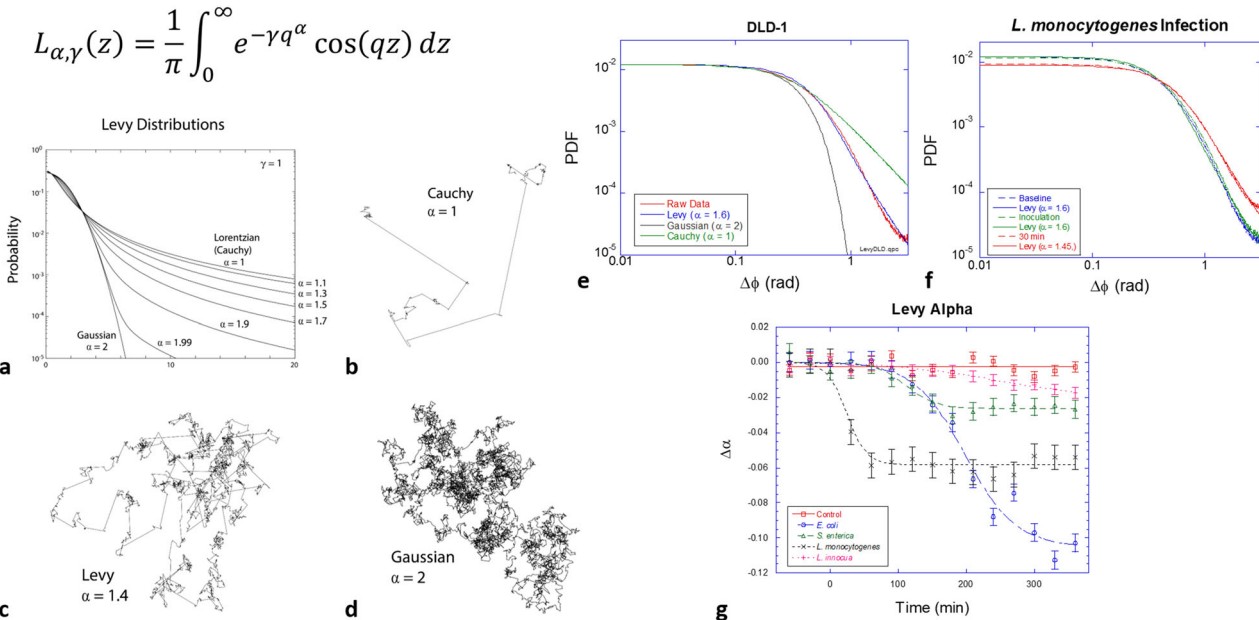

**Fig. 5 Lévy statistics in living tissue and anomalous random walks associated with Levy flights. a** Lévy probabilities. Lévy processes with smaller alpha have more frequent ballistic motions and are less ballistic with larger alpha. **b–d** Examples of Lévy random walks displaying the heavy tail of persistence lengths. **e** The PDF of DLD-1 tissue with a Lévy exponent of $\alpha = 1.6$. **f** The PDF of tissue infected by *L. monocytogenes* ($10^7$ CFU/well), showing a measurable shift to $\alpha = 1.4$ in only 1.5 h. **g** Temporal shifts of Levy exponents from multiple samples infected with the four strains of bacteria. Error bars represent standard error. A logistic function was used to fit the data.

ability of these bacteria to invade and colonize volumetric tissue. *E. coli* shows only slightly increased activity in the core region over long times, and the dominant blue shifts are restricted to the outer region, consistent with the weaker ability of *E. coli* to penetrate the volume of living tissue. The non-invasive (and non-pathogenic) strain *L. innocua* is indistinguishable from the control (no infection). The shifts begin in the outer regions of the spheroid and move inward over several hours as the infection spreads, as shown in Fig. 4c for the profiles of the average spectral shift, with redshifts in the inner core and blue shifts in the outer shell that move inward over hours.

**Lévy-alpha spectroscopy.** Lévy-stable probability distributions (shown in Fig. 5a) have a central role in probability theory and are being applied with increasing regularity to physical systems that sample from distributions with heavy tails[48]. For instance, Lévy walks (Fig. 5b–d) with Lévy distributions of path lengths or persistence times are random walks that have divergent mean-squared displacement and have been found to describe diverse processes in the biological sciences[48], including foraging behavior that may benefit from Lévy walks to optimize search strategies[9,49,50], bacterial motion[8,51], and the migration of T-cells[52]. Recent evidence suggests that Lévy walks govern intracellular motions[53–55]. This is of particular interest in our use of Doppler light scattering to interrogate living tissue. We have developed a new measure of the dynamic response of living tissue based on phase-sensitive BDI and probability distribution functions (PDF) of phase changes in 40-millisecond windows (see "Methods" section). The chief parameter for a symmetric Lévy distribution is known as the "alpha" parameter. When $\alpha = 2$, then the Lévy distribution is identical to a Gaussian with convergent moments. When $\alpha = 1$, the Lévy distribution is identical to a Lorentzian lineshape, also known as a Cauchy distribution. The first moments diverge for all $\alpha < 2$ and diverge more rapidly as alpha approaches the Cauchy distribution with the heaviest tails. Heavy tails mean strong outliers of large and rare events. The PDF of the phase fluctuations in healthy

DLD-1 tissue is shown in Fig. 5e with best fits to Gaussian ($\alpha = 2$) and Cauchy ($\alpha = 1$) distributions and a Lévy distribution with $\alpha = 1.6$ (best fit), showing clear evidence for the collective Lévy properties of the intracellular dynamics of the DLD-1 tissue sample (Supplementary Fig. 5).

Levy flights enable a "Levy alpha spectroscopy" of bacterial infection based on phase-sensitive BDI by including the effect of bacterial infection on the Lévy-alpha exponent. An example is shown in Fig. 5f for DLD-1 tumor spheroids infected by *L. monocytogenes*. Within only 30 m after infection, a measurable change in the PDF has occurred. The original PDF before the infection has a Levy distribution with a Levy exponent $\alpha = 1.6$, which reduces to $\alpha = 1.4$ (more "Cauchy-like") after infection. Lower alpha values pertain to heavier tails and stronger outliers in the PDF, possibly related to increased transport persistence length caused by the internal propulsion of the pathogenic bacteria. The change in alpha values for multiple sample replicates responding to the four bacterial strains studied here are shown in Fig. 5g. The rapid shift of the Levy alpha for the case of *L. monocytogenes* inoculation is consistent with rapid invasion by *L. monocytogenes*[6,7,14]. Other strains show a slower reduction in the Levy alpha. The largest decrease is observed for *E. coli*, which also had a large increase in the backscatter brightness for this bacterium at $10^7$ CFU/well.

**Antibiotic response of bacterial infection.** An important application of the living tissue sentinels is to serve as a dynamic substrate for the study of antibiotic sensitivity and resistance. Previous studies of BDI in drug development and personalized medicine classified the response of living biopsy samples from patients enrolled in clinical trials into sensitive or resistant cohorts[22]. In the current microbiology context, the test samples are the immortalized DLD-1 sentinels that perform as dynamic substrates on which to observe the dynamic effects of infection and to observe how the infection responds to antibiotic treatments. The goal is to identify biodynamic spectral signatures that correlate with the efficacy of the antibiotic.

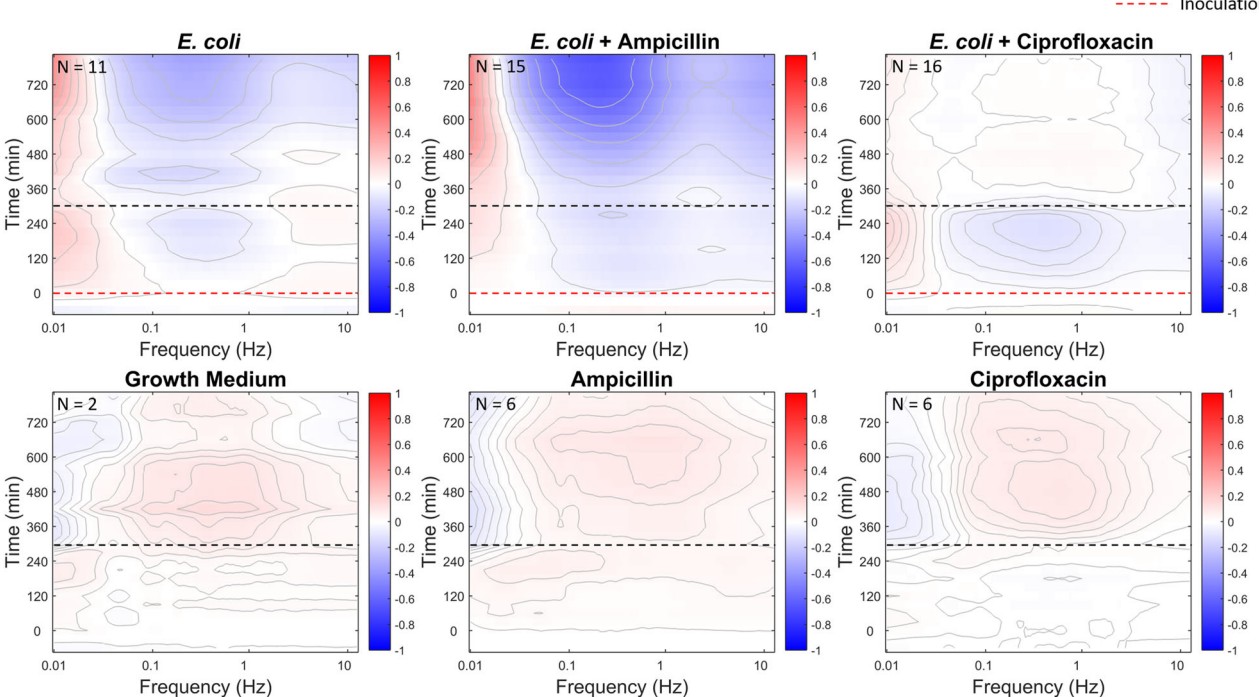

**Fig. 6 Time evolution of the spectral density of DLD-1 sentinels.** The top row shows the (intensity normalized) spectrogram responses for the infected DLD-1 sentinels. The dashed red line is the time of bacterial inoculation, and the dashed black line is the time of antibiotic application. The bottom row shows the responses of the DLD-1 sentinel controls without bacterial infection. *E. coli* infection induces broadband suppression of activity (top of left column). Treatment with ampicillin (middle column) has little effect on the infected tissue, but ciprofloxacin (right column) halts and reverses the infection, returning the tissue to a response similar to the uninfected control. The replicate numbers are shown at the top-left corner of the spectrogram plots.

An example of antibiotic treatment is shown in Fig. 6 for *E. coli* at an initial load of $10^6$ CFU per well infecting DLD-1 tissue sentinels and then treated with ampicillin and ciprofloxacin (broadband antibiotics). The *E. coli* strain used for the antibiotic resistance assay has ampicillin resistance enabling the bacteria to survive up to a dose of 50 μg/ml ampicillin[43,56]. In contrast, ciprofloxacin suppresses the proliferation of *E. coli* because ampicillin resistance is unrelated to ciprofloxacin resistance[57]. At the time of inoculation, the DLD-1 sentinels are exposed to $10^6$ CFU per well of *E. coli*. BDI tracks the *E. coli* infection spectrum of DLD-1 for 4.5 h, at which time three different treatments are applied to the DLD-1 sentinels in separate wells: growth medium (control), ampicillin (50 μg/ml), and ciprofloxacin (3.5 μg/ml)[56]. The time-course spectrograms are shown in Fig. 6 for the three treatments. The top row shows the responses for the infected DLD-1 sentinels. The dashed red line is the time of bacterial inoculation, and the dashed black line is the time of antibiotic application. The bottom row shows the responses of the DLD-1 sentinel controls without bacterial infection. The controls are similar for all treatments, displaying a mild mid-frequency enhancement in response to the treatment that includes replenished growth medium with nutrients. The DLD-1 sentinels with bacterial infection show a broadband suppression of activity within the tissue. Application of ampicillin has little effect (for this ampicillin-resistant strain), while ciprofloxacin removes the bacterial infection and returns the tissue to a condition comparable to the control.

## Discussion

Light scattered from living tissue displays a broad range of Doppler frequency shifts related to complex cellular processes and their associated dynamic motion. The Doppler fingerprint of

living tissue is extremely sensitive to subtle changes in intracellular dynamics, and BDI provides a powerful new technique for monitoring the response of 3D living tissue to xenobiotic challenges. In this paper, we describe the first use of BDI to monitor the infection of 3D living tissue by bacteria. Bacteria affect many of the dynamic processes within the living host, allowing the cellular response to perform the role of a living sentinel, reporting on the effects of the bacterial infection as well as monitoring the efficacy of antibiotic treatments. To illustrate the principle of the living sentinel, tumor spheroids of the DLD-1 colon adenocarcinoma cell lines were used to highlight different characteristics caused by infection by different bacterial strains. The spectral enhancements represent changes in dynamics with different frequency ranges associated with different types of intracellular motion. In particular, the Levy alpha spectroscopy represents a new form of an optical assay that can assess the character of bacterial infections. This work demonstrates the potential to translate BDI to the clinic to test for antibiotic-resistant infections.

The in vitro protocols presented here could perform a personalized selection of antibiotics by incubating patient-derived bacteria with the tissue sentinels and then testing the efficacy of a battery of conventional antibiotics. Many existing assays for antibiotic selection require ~6–8 h[58] with recent rapid assays able to provide results within 3 h[59]. Scaling the size of the current 600-micron scale tissue sentinels down to tissue microclusters that contain approximately a thousand cells could be sensitive enough to measure infection of a microcluster by a single bacterium once it has doubled ten times over about 3 h, making the biodynamic approach comparable to other rapid assays in terms of speed.

All measurements presented here were performed in vitro on artificially grown tissue sentinels. An intriguing prospect is to perform biodynamic imaging in vivo. The current optical design

uses a Mach–Zehnder interferometer that is sensitive to mechanical vibrations and patient motions, limiting its use in the in vivo setting. However, merging biodynamic imaging with common-path configurations[60,61] could remove sensitivity to global motions caused by breathing and by diastolic and systolic pressures. This raises the possibility to perform in vivo scans of the intracellular activity of infected tissue on accessible external surfaces or on internal surfaces through the use of an endoscope.

## Methods

**Eukaryotic and prokaryotic sample preparation**. Bacteria samples were cultured in LB (Lysogeny-broth) medium for 24 h at 37 °C. LB medium is a solution of 1% tryptone, 0.5% bacto-yeast extract, 1% sodium chloride and 0.1% glucose. After 24-hours, bacterial concentration in the LB medium reaches $10^8$ CFU/ml. The replication speed of *L. innocua* is slower than either *E. coli* GFP, *S. enterica*, or *L. monocytogenes*, requiring a 48-h culturing period.

DLD-1 is an adenocarcinoma standard colon cancer cell line. DLD-1 spheroids were chosen for studying bacteria-mammalian cell interaction because food-borne pathogens interact preferentially with epithelial cells, and the properties of the DLD-1 have been studied thoroughly by BDI. The DLD-1 cell stock was purchased from American Type Culture Collection (ATCC, Manassas VA) and cells were grown in RPMI-1640 medium containing 25 mm HEPES buffer (Gibco) supplemented with 10% fetal bovine serum (Atlanta Biologicals) and 100 U penicillin/ml-100 µg/ml streptomycin (Gibco). Multicellular spheroids were created by seeding 10,000 cells per well into spheroid microplates (Corning) for 4–5 days with media being replenished every 2 days. For biodynamic experiments, the spheroids were transferred to 96-well BioCoat plates (Corning) in an antibiotic-free growth medium and allowed to adhere for several hours. Paclitaxel was purchased from Selleck Chemicals.

Desired bacterial doses ($10^7$ CFU/well) applied to the DLD samples were achieved by diluting the bacterial medium with a mammalian cell-culturing medium (RPMI-1640). To minimize the pipetting perturbation on DLD-1 cells, the volume of the applied medium was set to 10 µl. To apply $10^6$ CFU, the prepared bacterial medium was diluted to 10%.

Experiments consisted of 16-well measurements on a 96-well plate. Each well contained a single DLD-1 spheroid and was filled with a growth medium volume of 350 µl. DLD-1 spheroids were immobilized on the bottom of the wells using poly-D-lysin. To infect the spheroids, the DLD-1 samples were exposed to the diluted bacteria medium allowing the bacteria to interact with the cells directly. Before applying bacteria, 3 baseline measurements were made over 90 m. After the baseline measurement, 10 µl of the diluted bacterial medium was pipetted into wells. Each well was monitored repeatedly by BDI for more than 6 h to detect interactions between bacteria and DLD-1 cells.

Bacterial antibiotic responses were measured by applying antibiotics after applying bacteria to the DLD-1 spheroids. To measure antibiotic resistance, *E. coli* green-fluorescence protein (GFP) strain was selected. *E. coli* GFP has ampicillin resistance because genes are genetically engineered to have an ampicillin resistance gene at the same promoter as the GFP-synthesizing gene[43]. Ampicillin and ciprofloxacin were prepared with doses of 50 and 3.5 µg/ml, respectively. Ampicillin and ciprofloxacin are broadband antibiotics, which are effective on gram-negative and gram-positive strains. *E. coli* GFP was applied to the DLD-1 spheroids. The antibiotics were applied 4.5 h after the infection, after which the Doppler spectra were measured for up to 9 h.

**Ranging with biodynamic imaging**. Biodynamic imaging (BDI) is configured as a Mach–Zehnder digital holographic interferometer with a low-coherence super-luminescent-diode (SLD) infrared light source (Fig. 1a). The wavelength of the SLD used in these experiments is 840 nm with 50 nm bandwidth, and the estimated coherence length is around 10 µm. Interference only occurs when the optical path length difference between the two arms is within the coherence length. Coherence-gating uses the optical path-sensitive interference characteristics of the low-coherence to select photons scattered from a specified distance inside the tissue.

Tumor spheroids have three-dimensional isotropic dynamics and BDI measures the volumetric dynamics of tumor spheroids. The coherence gate selects a two-dimensional cross-sectional area of the spheroids. When an interference pattern forms at the Fourier plane in Fig. 1, an optical coherence image (OCI) is acquired by performing a spatial Fourier transform of the digital hologram. A time series of multiple OCI frames represents the time-dependent fluctuating dynamic speckle carrying broadband Doppler signals that encode the cellular dynamics. A Doppler shift induced by scattering with a dynamic particle is given by [21],

$$\omega_D = v \frac{4n\pi}{\lambda} \sin\frac{\theta}{2} \cos\xi, \qquad (2)$$

where $v$, $n$, $\theta$, and $\xi$ are the speed, refractive index of the medium, scattering angle, and angle between the velocity of the particle and wavevector, respectively. As implemented here, BDI captures the backscatter signals and the 3-dimensional dynamics are assumed to be isotropic. Therefore the scattering angle $\theta$ is $\pi$ and $0 \leq \xi \leq 2\pi$ with an isotropic distribution. Due to the isotropic dynamic distribution, the

Doppler signal has a broadband power spectral density with zero mean, and the variance depends on $\omega_D^2$. Doppler power spectra of the low-frequency band from 0.01 to 0.1 Hz corresponds to speeds of 3–30 nm/s relating to cell shape changes and slowly-spreading motions such as cell crawling. The mid-frequency band from 0.1 to 1 Hz corresponds to speeds from 30 to 300 nm/s associated with nuclear motions and membrane processes. The high-frequency band from 1 to 10 Hz corresponds to speeds from 0.3 to 3 µm/s associated with organelle motion[35]. Power spectra are acquired by performing a temporal Fourier transform of the OCI reconstructions for individual pixels and then averaging over the image pixels. The BDI system records digital holographic images at 25 frames-per-second (fps). Each measurement of a well records 10-background images and 2048 holographic images for 85 s. A full-cycle scan of 16 wells in the multi-well plate takes 30 min.

**Time-lapse TDSI**. Tissue-dynamics spectroscopic imaging (TDSI) maps the spatial response across a tissue sample[39]. Although tissues are grown from genetically identical cells, their interactions with the external environment are not homogeneous. Conventional BDI takes averages of the power spectra across the entire tissue sample, but spatial information is lost. To study the spatial characteristics of tissues, the spectral responses are obtained for each pixel of the OCI reconstruction. After measuring the baseline spectra for each pixel, the TDSI algorithm compares the spectral variation over time compared to the averaged baseline of the sample. Spectral shape changes can be mapped using biomarkers, and the multi-color algorithm marks pixels with red when the spectral response is strongly correlated to the biomarker and blue for anticorrelated. The TDSI biomarker used in Fig. 4 is SDIP0 that represents a blue or redshift of frequencies caused by increased or decreased speeds of intracellular motion (Supplementary Note 3 and Tables 3–5).

**Phase-sensitive BDI**. A reconstructed image contains phase information through the Fourier transform of the digital hologram that generates a complex-valued reconstructed image. In the conventional operation mode of BDI, the Doppler signals from dynamic speckle are extracted by taking the absolute value of the reconstructed image. This is equivalent to homodyne detection in which the fluctuation power spectrum is the superposition among all the different Doppler frequency shifts. Homodyne fluctuation spectra are stabilized against phase drift of the Mach–Zehnder interferometric configuration of the optical system, which is sensitive to low-frequency mechanical disturbances. In phase-sensitive BDI the complex reconstructed image is retained and the phase for each pixel is calculated using a 4-quadrant arctangent. However, by taking phase differences between frames (40 ms), global phase drift is subtracted, and conventional phase unwrapping techniques are applied at the branch cut. Histogramming the phase differences generates a phase probability distribution function (PDF), which is analyzed for Lévy-stable properties. The tail of the Lévy distribution of phase differences is insensitive to global phase drift.

**Lévy-alpha spectroscopy**. The phase difference $\Delta\phi$ distribution in Fig. 5e, f shows a heavy tail. If dynamic particles move as normal random walkers the distribution would be a Gaussian distribution. However, many biological processes have random walks with anomalous (and infrequent) ballistic motions known as a Lévy walk. Lévy walk displacement ($\Delta x$) distributions have power-law tails. The slope of the tail is determined by the parameter α of the Lévy distribution, which is given by ref. [62] as

$$L_{\alpha,\gamma}(z) = \frac{1}{\pi}\int_0^\infty e^{-\gamma q^\alpha} \cos(qz)\mathrm{d}q. \qquad (3)$$

Lévy distributions obtained numerically are used to fit the phase difference distribution of tumor spheroids with a residual-square minimization algorithm.

**Power spectral density and spectrogram normalization methods**. When the backscatter brightness of a target changes over time, the un-normalized spectral power density includes the shift in optical power. However, it may be desirable to compensate for this changing brightness to focus on spectral shifts, and normalization choices emerge. Three different normalizations of time-frequency spectrograms are used.

$$S_{norm}(\nu, t) = S_{raw}(\nu, t), \qquad (4)$$

$$S_{norm}(\nu, t) = \frac{S_{raw}(\nu, t)}{I^2}, \qquad (5)$$

$$S_{norm}(\nu, t) = \frac{S_{raw}(\nu, t)}{\int_{0.01}^{12.5} S_{raw}(\nu, t)\mathrm{d}\nu}. \qquad (6)$$

The first method (in Eq. (4)) is un-normalized, capturing changes in sample brightness. The second method (in Eq. (5)) normalizes spectral density by the total signal intensity squared for the so-called intensity normalization. This normalization leads to time-frequency spectrograms that faithfully reflect the relative change in spectral power density across all frequencies. The third method (in Eq. (6)) normalizes the total power spectral density within the sampling bandwidth for the so-called zero-sum normalization. The sum of the spectral

changes across the BDI bandwidth 0.01–12.5 Hz is set to zero for each time frame. This normalization captures spectral shifts constrained by a zero-sum.

**Statistics and reproducibility**. All statistics in the manuscript were processed by MATLAB (R2015a). The statistical averages and standard deviations were acquired among all replicates without data rejections. The replicate numbers are defined as a number of independent specimens.

**Reporting summary**. Further information on research design is available in the Nature Research Reporting Summary linked to this article.

## Data availability

Data are available upon request mailed to nolte@purdue.edu

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

## Acknowledgements

This work was supported by NSF grant CBET-1911357.

## Author contributions

H.C., Z.H., and D.N. conducted BDI measurements and the spectral phenotype analysis. Z.L. and D.N. developed TDSI algorithms and performed the regional speckle phenotype analysis. H.C., J.Z., E.X., and M.L. prepared bacterial samples. J.T. designed host-cell characteristics and prepared DLD-1 spheroids. H.C., J.Z., E.X., M.L., and D. N. contributed to measuring pathogenicity and antibiotic resistance bacteria spectral phenotype measurements. All authors contributed to the discussions and writing of the manuscript.

## Competing interests

David Nolte and John Turek have a financial interest in Animated Dynamics, Inc. that is commercializing cancer therapy selection services using biodynamic imaging systems. The remaining authors declare no competing interests.
