## [Peer Review File · Communications Biology]

Reviewers' comments:

Reviewer #1 (Remarks to the Author):

See attached file

Reviewer #2 (Remarks to the Author):

COMMSBIO-20-0954-T, "Doppler Imaging Bacterial Infection of Living Tissue Sentinels," by H. Choi et al.

This manuscript presents a study on using dynamic motion within 3D tissue cultures to characterize bacterial infection. The approach uses phase sensitive measurements to infer Doppler shifts to detect motions in the low frequency range between 10 mHz and 10 Hz. The approach is applied to examine cultures of immortalized cells which have been exposed to different bacterial species. The approach is aimed at serving as an assay of bacterial growth for the purpose of assessing antibiotic resistance.

Biodynamic imaging, the approach used here has been well characterized in the literature and applied in several areas. Novelty here is evaluation of the approach as a label free alternative to fluorescence assays for monitoring pathogens. Significantly, BDI has the stability to detect deep sub-Hertz Doppler shifts which are difficult to detect via fluorescence due to photobleaching and drifting baselines. The manuscript could use a better explanation of how the technique offers stability and how the key metrics are calculated (Line 135-149). The description of the spectrograms in the subsequent paragraph is much clearer.

The spatial maps of dynamics within the cell sample is unique for this approach and can provide insight into bacterial invasion. The measurements in Fig 4 show modulation during inoculation. However, these appear to be single measurements and replicates should be presented to verify these effects.

The use of Levy statistics in an interesting approach for analyzing the fluctuation information from the samples. The model includes more variation than usual random walks where large sporadic deviations are included. The BDI are analyzed using this model and shows variation due to infection. A link is drawn between these dynamics and the motility of E. Coli. This is a very interesting link that should be examined in greater detail and/or confirmed with other methods.

The spectrogram analysis is used to examine modulation due to antibiotic application. These are quite convincing that the this method can detect changes in dynamics. However, again it is not clear if these are single measurements and how well these results stand up to replication. In summary, this is an interesting new application for a technique that has been previously developed. The approach shows potential for characterizing infection in 3D cell cultures and modulation has been observed due to antibiotic treatment. This work should be published after addressing the concerns listed above.

Reviewer #3 (Remarks to the Author):

Overall, this is a very intriguing paper that both introduces a new technique for quantitatively monitoring characteristics of bacterial invasion and applies the technology to study antibiotic resistance, which is a critical area of unmet need. I believe that this manuscript merits publication in Communications Biology provided that the following comments are addressed.

Major Comments:

(1) Since these spheroids are derived from adenocarcinoma cells and not normal cells, would the authors expect the bacteria to behave the same in normal tissue as they were observed to behave in the tumor spheroids in this study? Fig. S2 suggests that some of the measured parameters are

even different for tumor spheroids that are grown in different ways, so the measured parameters may behave quite differently in normal tissue.

(2) How much of this work is translational to in vivo settings? What aspects of the current setup would need to be changed in order to translate this work in vivo, and how challenging do the authors anticipate these changes to be? How much are the measured parameters expected to change in vivo relative to in tumor spheroids? Are the measured parameters likely to behave very differently in vivo, considering that they even behaved differently for different in vitro spheroid growth methods? (Fig. S2)

(3) Could the antibiotic resistance work be used to provide an earlier prognosis of how the bacteria are responding to a given treatment so that the treatment can be changed as soon as possible if resistance to a certain drug is observed?

(4) Fig. 1(b,d,e): How do the bacteria alter the speckle pattern? Is it due to their motion or is there some other way that they change the interference pattern?

(5) Fig. 2: How is the dynamic range of the spectral density calculated, and what does it signify physically/physiologically?

(6) Lines 182-191 and 523-528: How do the authors know which frequency bands are associated with which cellular properties? I feel that this merits a bit more explanation, as well as some citations if available.

(7) Fig. 5: What is the physical interpretation for the change in the alpha parameter, as it relates to bacterial behavior?

Minor Comments:

Fig. 1(C): What is the significance of the region of the image that has the red box over it? It would help to expand the caption to explain that in further detail.

Line 160: "monotonically" is misspelled

Fig. 2a,b: Are these images of the entire spheroid, or have they been cropped/segmented in some way?

Fig. 3: The caption mentions *S. enterica* but I do not see that bacterium labeled in any of the figure panels.

Fig. 4c: What are the units of spectral shift?

Responses to Reviewers:

Reviewer #1

1. *“Several terminologies are used in the manuscript without clear scientific definition and rigor like “Doppler light scattering” and “phase-sensitive Doppler shift”.”*

We have added the following text at Line 37:

Dynamic speckle detected using BDI is caused by light scattering from moving intracellular components that have persistent transport that places them in the Doppler light scattering regime for which the persistence length of the transport is greater than the reduced wavelength of the probe light, expressed as $\ell_p > \lambda_0 / 2n$, where ℓ_p is the mean transport persistence length, $\lambda_0 = 840$ nm is the free-space wavelength and $n \approx 1.4$ is the refractive medium of the tissue. (For a detailed analysis of intracellular Doppler light scattering see Ref. [19].)

2. *“What is the benefit of low coherence interferometry”*

We have added the following text at Line 35:

Biodynamic imaging [16] is based on low-coherence interferometry with coherence-gated ranging based on digital holography to select scattered light from a specific optical depth inside living tissue while rejecting incoherent background [10, 17, 18].

3. – 7. *Specific suggestions regarding figure formats and figure captions*

We have addressed all of the suggestions regarding figure formats and captions.

Reviewer #2

1. *“Better explanation of how the technique offers stability and how the key metrics are calculated (Line 135 – 149)”*

We have changed the description of phase-sensitive BDI at Line 615:

A reconstructed image contains phase information through the Fourier transform of the digital hologram that generates a complex-valued reconstructed image. In the conventional operation mode of BDI the Doppler signals from dynamic speckle are extracted by taking the absolute value of the reconstructed image. This is equivalent to homodyne detection in which the fluctuation power spectrum is the superposition among all the different Doppler frequency shifts. Homodyne fluctuation spectra are stabilized against phase drift of the Mach-Zehnder interferometric configuration of the optical system which is sensitive to low-frequency mechanical disturbances. In phase-sensitive BDI the complex reconstructed image is retained and the phase for each pixel is calculated using a 4-quadrant arctangent, but it is subject to drift in the global phase. However, by taking phase differences between frames (40 msec), the global phase drift is subtracted, and conventional phase unwrapping techniques are applied at the branch cut. Histogramming

the phase differences generates a phase probability distribution function (PDF) which is analyzed for Lévy-stable properties. The tail of the Lévy distribution of phase differences is insensitive to global phase drift.

In addition, a detailed explanation of the key metrics of biodynamic imaging has been added as **Section S6** in the supplemental material document.

2. *Fig. 4 Replicates*

Tissue dynamics spectroscopic imaging (TDSI) was performed on 12 replicates that all showed qualitatively similar behavior. The blue shift has slightly differing asymmetry in some of the samples, a was slightly weaker on other samples. The data shown in Fig. 4 are “representative” of the types of spatial variation that were observed.

3. *“A link is drawn between these dynamics and the motility of E. coli. This is a very interesting link that should be examined in greater detail and/or confirmed with other methods.”*

At this time, we are unable to link the strong decrease of the Levy alpha under *E. coli* infection to direct motility properties of the bacterium. Therefore, we removed the speculation that was in the original manuscript while still describing the observation (averaged over 15 replicates with small variance) to keep it open for others to explore. This will be a topic of future research.

4. *“These are quite convincing that this method can detect changes in dynamics. However, again it is not clear if these are single measurements and how well these results stand up to replication.”*

All experiments were performed with a large number of replicates, usually greater than 12 for each condition. Each “experiment” was performed with typically 3 to 6 replicates per condition per plate and repeated across typically 5 plates spanning several months. We now point this out where appropriate in the revised manuscript.

Reviewer #3

1. *“would the authors expect the bacteria to behave the same in normal tissue as they were observed to behave in the tumor spheroids in this study? Fig. S2 suggests that some of the measured parameters are even different for tumor spheroids that are grown in different ways, so the measured parameters may behave quite differently in normal tissue.”*

Cells that can be continuously cultured to form 3D structures (spheroids or organoids) are altered by either genetic mutation or a virus (retrovirus). *Normal*, non-transformed, primary cells cannot be used for long term studies because the phenotype typically changes with each cell passage and is not stable. Therefore, there are no truly normal cell lines (without genetic mutations or not virally transformed) that can be continuously passaged to form 3D structures that have a stable phenotype.

2. Translation and challenges to the *in vivo* setting.

We have added the following text at Line 375:

All measurements presented here were performed *in vitro* on artificially-grown tissue sentinels. An intriguing prospect is to perform biodynamic imaging *in vivo*. The current optical design uses a Mach-Zehnder interferometer that is sensitive to mechanical vibrations and patient motions, limiting its use in the *in vivo* setting. However, merging biodynamic imaging with common-path configurations [59, 60] could remove sensitivity to global motions caused by breathing and by diastolic and systolic pressures. This raises the possibility to perform *in vivo* scans of the intracellular activity of infected tissue on accessible external surfaces or on internal surfaces through the use of an endoscope.

3. Earlier prognosis of resistance

We have added the following text at Line 367:

The *in vitro* protocols presented here could perform personalized selection of antibiotics by incubating patient-derived bacteria with the tissue sentinels and then testing the efficacy of a battery of conventional antibiotics. Many existing assays for antibiotic selection require approximately 6 – 8 hours [57] with recent rapid assays able to provide results within 3 hours [58]. Scaling the size of the current 600-micron scale tissue sentinels down to tissue microclusters that contain approximately a thousand cells could be sensitive enough to measure infection of a microcluster by a single bacterium once it has doubled ten times over about 3 hours, making the biodynamic approach comparable to other rapid assays in terms of speed.

4. “Fig. 1 How do the bacteria alter the speckle pattern?”

In the figure caption we point the reader to the fourth image column, which is the motility map.

5. Fig. 2 Dynamic range

We have added the following text at Line 153:

The dynamic range is defined as the ratio of the spectral amplitude at the lowest frequency (10 mHz) to the spectral amplitude at the Nyquist frequency (12.5 Hz). The amplitude at the Nyquist frequency is also called the “Nyquist floor”, which is another useful dynamic marker. The Nyquist floor increases when there is an increase in high-speed transport, for instance caused by organelle or vesicle transport. The rise of the Nyquist floor decreases the dynamic range if there is no corresponding increase of amplitude at the lowest frequency. On the other hand, the dynamic range increases if the dynamics around 0.01 Hz in the low-frequency band contains an increased fraction of intracellular components that move slowly. The DR is calculated by averaging the first 5 spectral components and the last 5 spectral components.

6. *“Lines 182-191 and 523-528: How do the authors know which frequency bands are associated with which cellular properties? I feel that this merits a bit more explanation, as well as some citations if available.”*

We have added the following text at Line 61:

The speeds of intracellular dynamics range across three orders of magnitude from tens of nanometers per second (cell crawling, metastasis, blebbing, apoptotic bodies) [28-31] to tens of microns per second (organelles, vesicles, mitochondria) [32-35] with associated Doppler frequencies spanning from 10 mHz to 10 Hz which correspond to intracellular speeds from 3 nm/s to 3 μ m/s [19]. The conversion from intracellular speed to Doppler frequency for the infrared backscatter geometry is $\Delta\omega_D = \vec{q} \cdot \vec{v}$, where $q = 4\pi n / \lambda_0$ is the momentum transfer and v is the intracellular speed.

7. *“Fig. 5: What is the physical interpretation for the change in the alpha parameter, as it relates to bacterial behavior?”*

We have added the following text at Line 279:

The chief parameter for a symmetric Lévy distribution is known as the “alpha” parameter. When $\alpha = 2$, then the Lévy distribution is identical to a Gaussian with convergent moments. When $\alpha = 1$, the Lévy distribution is identical to a Lorentzian lineshape, also known as a Cauchy distribution. The first moments diverge for all $\alpha < 2$ and diverge more rapidly as alpha approaches the Cauchy distribution with the heaviest tails. Heavy tails mean strong outliers of large and rare events.

And a sentence at Line 304:

Lower alpha values pertain to heavier tails and stronger outliers in the PDF, possibly related to increased transport persistence length caused by the internal propulsion of the pathogenic bacteria.

We also performed a new set of experiments to confirm that drugs that inhibit intracellular dynamics increased to alpha value (reduced the tail of long-persistence transport). A new section **Section S5** has been added to the supplemental information document at Line 136.

8. *Minor comments, corrections and formatting:*

We have addressed all the minor comments, corrections and formatting.

REVIEWERS' COMMENTS:

Reviewer #1 (Remarks to the Author):

The revised manuscript has addressed most of my concerns.

1. "Doppler light scattering" is a jargon only used by the authors here and in their other paper. It's better to give a definition.
2. It will be better to show the std in Fig.2c, which will be clearer for the readers. "N" is explained in the caption of Fig. 3 not Fig. 2.

Reviewer #3 (Remarks to the Author):

The authors have appropriately addressed all of my previous comments and I believe that this manuscript is ready for publication.

Final changes made at the request of Reviewer 1:

1. "Doppler light scattering" is a jargon only used by the authors here and in their other paper. It's better to give a definition. - We have addressed the referee #1's remark by adding the definition of the Doppler light scattering in the text at Line 46.

Dynamic speckle detected using BDI is caused by light scattering from moving intracellular components that have persistent transport that places them in the Doppler light scattering regime, based on Doppler shifts caused by dynamic light scattering [19], for which the persistence length of the transport is greater than the reduced wavelength of the probe light, expressed as $\ell_p > \lambda_0 / 2n$, where ℓ_p is the mean transport persistence length, $\lambda_0 = 840$ nm is the free-space wavelength and $n \approx 1.4$ is the refractive medium of the tissue.

2. It will be better to show the std in Fig.2c, which will be clearer for the readers. "N" is explained in the caption of Fig. 3 not Fig. 2. – We have added the error bars to Fig. 2c and d as requested. In Figure 3 caption, we described the meaning of replicate number N.